

# Seasonal changes in fish assemblage structure at a shallow seamount in the Gulf of California

Salvador J. Jorgensen[1,2], A. Peter Klimley[2], Arturo Muhlia-Melo[3] and Steven G. Morgan[4]

[1] Conservation Research Department, Monterey Bay Aquarium, Monterey, California, United States
[2] Department of Wildlife, Fish, & Conservation Biology, University of California, Davis, California, United States
[3] Fisheries Ecology Program, Centro de Investigaciones Biológicas del Noroeste S. C., La Paz, Baja California Sur, Mexico
[4] Bodega Marine Lab, University of California, Davis, Bodega Bay, California, United States

Corresponding author
Salvador J. Jorgensen,
sjorgensen@mbayaq.org

## ABSTRACT

Seamounts have generally been identified as locations that can promote elevated productivity, biomass and predator biodiversity. These properties attract seamount-associated fisheries where elevated harvests can be obtained relative to surrounding areas. There exists large variation in the geological and oceanographic environment among the thousands of locations that fall within the broad definition of seamount. Global seamount surveys have revealed that not all seamounts are hotspots of biodiversity, and there remains a strong need to understand the mechanisms that underlie variation in species richness observed. We examined the process of fish species assembly at El Bajo Espiritu Santo (EBES) seamount in the Gulf of California over a five-year study period. To effectively quantify the relative abundance of fast-moving and schooling fishes in a 'blue water' habitat, we developed a simplified underwater visual census (UVC) methodology and analysis framework suitable for this setting and applicable to future studies in similar environments. We found correlations between seasonally changing community structure and variability in oceanographic conditions. Individual species responses to thermal habitat at EBES revealed three distinct assemblages, a 'fall assemblage' tracking warmer overall temperature, a 'spring assemblage' correlated with cooler temperature, and a 'year-round assemblage' with no significant response to temperature. Species richness was greatest in spring, when cool and warm water masses stratified the water column and a greater number of species from all three assemblages co-occurred. We discuss our findings in the context of potential mechanisms that could account for predator biodiversity at shallow seamounts.

Subjects Biodiversity, Ecology, Marine Biology
Keywords Seamount, Marine ecology, Biodiversity, Underwater visual census, Marine biology, Community ecology, Oceanography, Fish assemblage, Gulf of California, Sea of Cortez

# INTRODUCTION

Seamounts have long been identified as important ocean habitats with elevated predator diversity (*Hubbs, 1959*; *Morato et al., 2010a*). Multi-species aggregations of fishes are

commonly reported at shallow seamounts (*Klimley & Butler, 1988*; *Rogers, 1994*; *Morato & Clark, 2007*). These aggregations are targeted by numerous fisheries and data suggest that catch rates for many fish species are higher near some seamounts relative to surrounding habitats (*Rogers, 1994*; *Genin, 2004*; *Morato et al., 2008*). Shallow seamounts consist of relatively shallow benthic habitat, within the euphotic zone, surrounded by adjacent deep-ocean (*Lueck & Mudge, 1997*; *Trasviña-Castro et al., 2003*; *Clark et al., 2010*; *Staudigel et al., 2010*). Thus, fishes aggregating at these sites are demersal and reef-associated, as well as pelagic species common in epipelagic environments (*Holland & Grubbs, 2007*; *Litvinov, 2007*; *Morato & Clark, 2007*; *Morato et al., 2008*). However, seamounts are highly variable in their geologic and oceanographic characteristics, and not all seamounts have fish aggregating properties (*Kvile et al., 2014*). Among the potential mechanisms and drivers, examining the oceanographic conditions promoting seamount productivity, predator aggregation, and species richness remains an important research gap (*Clark et al., 2012*; *McClain & Lundsten, 2015*; *Morato et al., 2015*). Tracking seamount community assembly over time in relation to natural oceanographic variability can provide unique insights into seamount ecology.

The mechanisms for why fishes sometimes occur in higher densities at shallow seamounts fall into two general categories. First, some seamounts may provide elevated foraging opportunities. Trophic subsidies likely result as seamounts generate conditions such as increased vertical nutrient fluxes and plankton retention that increase productivity and fuel higher trophic levels (*Lueck & Mudge, 1997*; *Genin, 2004*). Second, seamounts may provide spatial reference points or refugia for migratory species (*Klimley, 1993*; *Fréon & Dagorn, 2000*). However, the processes that determine the composition of fish species and elevated predator richness have received comparatively little attention, and remain uncertain (*McClain, 2007*; *Morato et al., 2010a*).

One hypothesis for increased fish species richness at seamounts is the enhanced availability of limiting trophic resources (*Worm, Lotze & Myers, 2003*; *McClain, 2007*; *Morato et al., 2010a*). This idea is essentially an extension of the 'species-energy' hypothesis (*Wright, 1983*), which predicts that the diversity of one trophic level is determined by the amount of energy available from the level below. If elevated forage availability at seamounts supports a greater number of individuals and, in turn, species richness, then food should be the primary resource of interest for fishes that visit seamounts, and more visitors (species) should occur when food supply is elevated.

An alternative hypothesis is that seamounts comprise diverse and heterogeneous habitats, which provide a variety of resources and environmental conditions suitable for a range of fish species and life history functions. The 'habitat heterogeneity' hypothesis is a long-standing tenet of terrestrial ecology (e.g. *MacArthur & Wilson, 1967*; *Tews et al., 2004*) whereby structurally complex habitats may provide more ways of exploiting environmental resources and thereby increase species diversity. Under this hypothesis, more species are predicted to co-occur when habitat is more complex, or when the breadth of resources exploited increases.

To explore processes of seamount community assembly and variation in fish species richness we tracked the relative abundance of conspicuous shallow seamount-associated

fish species at El Bajo Espiritu Santo (EBES) seamount over a five-year period. To overcome the difficulty in quantifying relative abundance for fast-moving and schooling fishes in a 'blue water' habitat, we developed a simplified underwater visual census (UVC) methodology and analysis framework suitable for this setting and applicable to future studies in similar environments. We compared the results of UVCs and experimental fishing surveys over time with oceanographic parameters to determine how changes in seamount community structure correlated with natural environmental variability. We discuss the results in the context of two hypotheses, 'species-energy' and 'habitat heterogeneity' as they relate to seamounts as predator diversity hotspots.

## METHODS

### Study area

EBES is located in the lower Gulf of California (24°42′N, 110°18′W) 56 km north of La Paz (Fig. 1A). The summit of the seamount reaches a minimum depth of 18 m and drops off to between approximately 500 and 1,000 m on all sides (for detailed bathymetric view see Fig. 1 in *Klimley, 1993*). The upper part of the seamount, within 30 m of the surface, forms a broad ridge that is approximately 500 m long and 100 m wide. Numerous demersal, reef-associated and pelagic fish species inhabited the water column between this ridge and the water surface, and this seamount-associated community was the focus our study.

### Underwater visual censuses

We conducted UVCs at EBES from 1999 to 2004 by measuring encounter rates for individuals of 27 conspicuous predator and prey fish species (Table 1) along transects using SCUBA. Divers recorded the identity and number of individuals of each target species encountered on a waterproof slate. Data were consistently collected by the same few divers throughout the study. Transects were initiated at random starting points near one end of the seamount ridge and proceeded on a compass bearing along its length in either a northwesterly or southeasterly direction. If currents were too strong to swim against, the census was initiated at a random starting point near the up-current end of the ridge, allowing divers to drift with the current. Censuses proceeded for 40 min while divers swam toward a constant compass bearing at a relatively constant speed of approximately 0.2 ms$^{-1}$. In order to sample across all depths between 0 and 30 m while maintaining safe SCUBA protocols, divers began sampling immediately upon leaving the surface while gradually descending. Once the maximum depth was reached (25–30 m), divers began a very gradual ascent with a goal of reaching the surface at the 40 min mark. Fish were only counted if they appeared within 5 m of the observer in any direction forward of a plane perpendicular to the swimming direction. Visibility was estimated by divers from the difference between the depth of the seamount substrate and the depth at which this substrate first became visible as the diver descended toward it. Visibility was measured only when a light patch of sand near the seamount summit (20 m) was not visible from the boat at the surface. During censuses, visibility ranged from 7 to > 20 m.

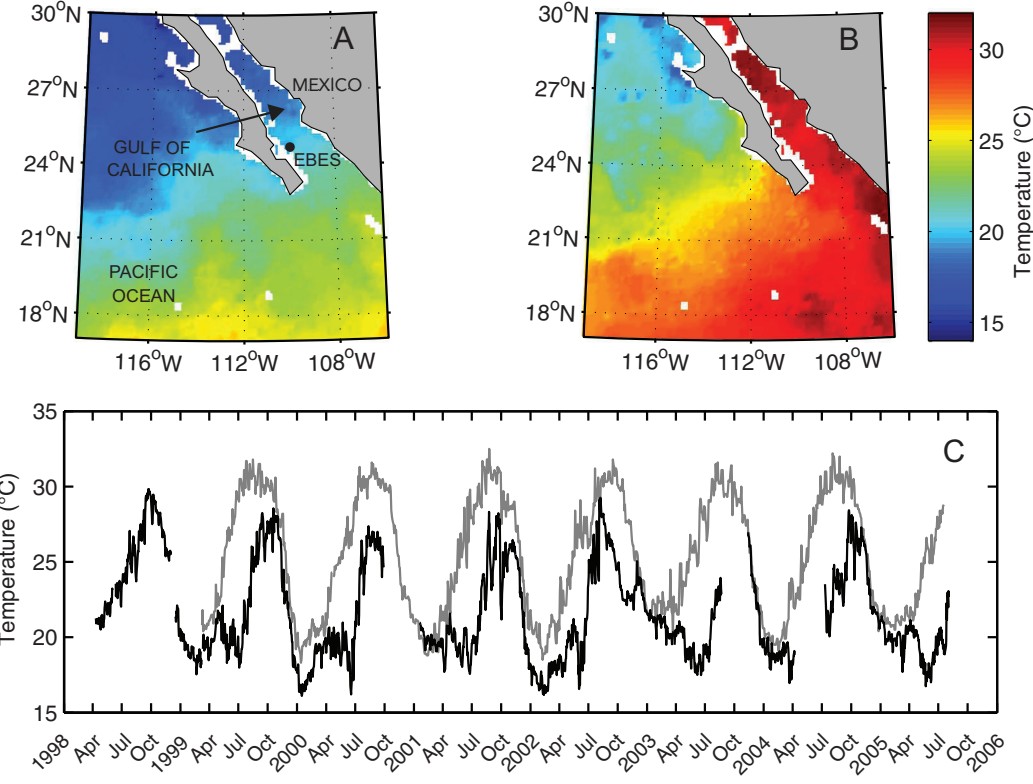

**Figure 1 Map of the Southern Gulf of California and Eastern Pacific Ocean overlain with NOAA AVHRR sea surface temperature (SST).** Images from (A) February 4, 2004, and (B) August 2, 2003 recorded, respectively during seasonal high and low peaks of the annual temperature cycle. (C) Sea surface temperature (gray line) and subsurface temperature (black line), measured at a depth of 30 m (T30), at EBES seamount (EBES; see filled circle in panel A).

Among species, considerable variability occurred in the number of individuals encountered per observation. For example, green jacks (*C. caballus*) often occurred in large schools (n > 128) when present at the seamount, while yellowtail (*S. lalandi*) counts seldom exceeded 12 individuals per transect. Additionally, several species were generally either absent from the seamount or present in large numbers. To minimize bias and sampling error, we organized our counts into classes or 'bins' with central values expressed on a $\log^2$ scale, i.e. centers 0 and $2^i$ where $i = 0$–7. For example, $2^0$ denotes a count of one individual, $2^1$ represents a class of between 1.4 and 2.8 individuals ($2^{0.5}$–$2^{1.5}$) with a central value of 2, and $2^2$ refers to a class of 2.9–5.6 individuals ($2^{1.5}$–$2^{2.5}$) with a central value of four, and so forth (Table 2). Thus, bin width expanded exponentially with the number of individuals counted assuring that even large numbers of rapidly moving fish could be accurately estimated to be within one of the range categories. When the number of individuals was too large or difficult to count accurately during censuses, the number was estimated to the nearest central value and corresponding ordinal score. This procedure provided an objective criterion for quickly sorting raw counts into a number of ordinal categories. Additionally this procedure resulted in a log-transformation of the data, thereby increasing the 'spread' and resolution for species with lower counts (see Tables 2 and 3).

**Table 1** List of fish species observed during underwater visual censuses at Espiritu Santo Seamount (EBES).

| Family or Class | Common name | Species | Environment* |
|---|---|---|---|
| Carangidae | Green jack | *Caranx caballus* | Pelagic-neritic |
| | Jack mackerel | *Trachurus symmetricus* | Pelagic-oceanic |
| | Mackerel scad | *Decapterus macarellus* | Pelagic-oceanic |
| | Yellowtail | *Seriola lalandi* | Benthopelagic |
| | Amberjack | *Seriola rivoliana* | Reef-associated |
| | Gafftopsail pompano | *Trachinotus rhodopus* | Reef-associated |
| Chondrichthyes | Hammerhead shark | *Sphyrna lewini* | Pelagic-oceanic |
| | Silky shark | *Carcharhinus falciformis* | Reef-associated |
| | Whale shark | *Rhincodon typus* | Pelagic-oceanic |
| | Manta | *Manta hamiltoni* | Reef-associated |
| Coryphaenidae | Dorado | *Coryphaena hippurus* | Pelagic-neritic |
| Istiophoridae | Sailfish | *Istiophorus platypterus* | Pelagic-oceanic |
| | Striped marlin | *Tetrapterus audax* | Pelagic-oceanic |
| | Blue marlin | *Makaira mazara* | Pelagic-oceanic |
| Lutjanidae | Red snapper | *Lutjanus peru* | Reef-associated |
| | Yellow snapper | *Lutjanus argentriventris* | Reef-associated |
| | Mullet snapper | *Lutjanus aratus* | Reef-associated |
| | Dog snapper | *Lutjanus novenfasciatus* | Reef-associated |
| | Colorado snapper | *Lutjanus colorado* | Reef-associated |
| | Barred snapper | *Hoplopagrus guntherii* | Reef-associated |
| | Rose-spotted snapper | *Lutjanus guttatus* | Reef-associated |
| Scombridae | Yellowfin tuna | *Thunnus albacares* | Pelagic-oceanic |
| | Black skipjack | *Euthynnus lineatus* | Pelagic-oceanic |
| | Wahoo | *Acanthocybium solandri* | Pelagic-oceanic |
| Serranidae | Creolfish | *Paranthias colonus* | Reef-associated |
| | Gulf grouper | *Mycteroperca jordani* | Reef-associated |
| | Leopard grouper | *Mycteroperca rosacea* | Reef-associated |

**Note:**
* Environment classification from FishBase (http://www.fishbase.org).

## Fishing surveys

After each census, we conducted standardized fishing surveys to detect species that were less frequently encountered while using SCUBA. We recorded captures (presence or absence) of *Thunnus albacares, Corphaena hippurus*, and *Acanthicybium solanrdi* during 60-min fishing periods. Our tackle consisted of monofilament line that was attached directly to a hook, and baited with a single herring, *Harengula thrissina*. Three of these 20-m long rigs were deployed simultaneously while the boat drifted over the seamount. Ethical review was not required for sampling of marine fishes in fishing surveys.

## Environmental records

At EBES, we recorded daily water temperature at a depth of 30 m using an in situ temperature logger (Onset Corporation, Tidbit Stowaway, and Water Temp Pro).

**Table 2 The bin centers and ranges corresponding to ordinal scores for underwater visual census counts.** Repeated censuses were averaged and then assigned ordinal scores according to the following $\log_2$ classification scheme.

| Ordinal score | Bin center | Bin range |
|---|---|---|
| 0 | 0 | 0 |
| 1 | $1\ (2^0)$ | $0–2^{0.5}$ |
| 2 | $2\ (2^1)$ | $2^{0.5}–2^{1.5}$ |
| 3 | $4\ (2^2)$ | $2^{1.5}–2^{2.5}$ |
| 4 | $8\ (2^3)$ | $2^{2.5}–2^{3.5}$ |
| 5 | $16\ (2^4)$ | $2^{3.5}–2^{4.5}$ |
| 6 | $32\ (2^5)$ | $2^{4.5}–2^{5.5}$ |
| 7 | $64\ (2^6)$ | $2^{5.5}–2^{6.5}$ |
| 8 | $128\ (2^7)$ | $> 2^{6.5}$ |

Sea surface temperature (SST14NA) was downloaded from the National Environmental Satellite, Data, and Information Service (NESDIS) online database (http://www.class. noaa.gov). This SST14NA product was generated every 48 h and referenced to in situ measurements at 1 m depth. Temperature profile were obtained opportunistically from casts taken at points along SW-NE transects that bisected EBES (*Trasviña-Castro et al., 2003*) on November 24, 1997, September 10, 1998, and June 22, 1999. Data collected $\leq 5$ km from the seamount (n = 4–7) were partitioned into depth bins and averaged for each date. A fourth temperature/depth profile was acquired from a data archiving tag (PAT; Wildlife Computers Inc.), which was attached to a shark (*Sphyrna lewini*) at EBES. Minimum and maximum temperatures were recorded from seven depths (4, 16, 100, 156, 260, 316, and 368 m) as the shark swam through the water column shortly after tagging on February 2, 2004. These data were later transmitted from the tag via satellite after the tag released from the shark (*Jorgensen, Klimley & Muhlia-Melo, 2009*).

## Analysis

Censuses were conducted at irregular intervals over the duration of the study period. Each independent observation was recorded as the mean value of replicate censuses (n = 1–4) that were completed within a calendar month. The mean number of encounters per transect for each species was then given an ordinal score following the $\log_2$ classification scheme (see Table 2). To examine the influence of seasonal oceanographic processes on the relative abundance of pelagic species at EBES, we used multiple logistic regression analysis (SAS Institute Inc., JMP). Of 27 species observed at the seamount, we considered only those present in > 10% of census observations (n = 17 species) for analysis. Ordinal score was regressed over monthly mean SST and monthly mean water temperature at 30 m depth (T30). We used the same technique for analyzing our fishing survey data, except the dependent variable was binary (presence and absence) rather than ordinal. To verify the assumption of temporal independence among observations, we performed time series analysis on the regression residuals. There was no significant autocorrelation at any lag time for any species ($\alpha = \pm 2$ SE). T30 measurements were not collected
**Table 3 Ordinal values from underwater visual counts of fishes at El Bajo Espiritu seamount from 1999–2004 by month.** Each value represents the mean number of individuals counted among replicate transects.

| | 1999 | | | 2000 | | | | 2001 | | 2002 | | | | | | 2003 | | | | | 2004 | | | | | |
|---|---|---|---|---|---|---|---|---|---|---|---|---|---|---|---|---|---|---|---|---|---|---|---|---|---|---|
| | May | Aug | Nov | Feb | Apr | Jun | Sep | Mar | Jul | Jun | Jul | Sep | Oct | Nov | Dec | Feb | Mar | May | Oct | Nov | Jan | Apr | May | Jun | Jul | Aug |
| *Caranx caballus* | 0 | 8 | 8 | 0 | 0 | 8 | 8 | 0 | 8 | 7 | 0 | 7 | 6 | 4 | 4 | 0 | 0 | 0 | 1 | 5 | 0 | 0 | 8 | 8 | 8 | 8 |
| *Decapterus macarellus* | 8 | 8 | 8 | 7 | 7 | 8 | 8 | 7 | 8 | 7 | 6 | 6 | 6 | 7 | 8 | 0 | 7 | 8 | 7 | 8 | 8 | 8 | 8 | 8 | 8 | 7 |
| *Euthynnus lineatus* | 8 | 8 | 8 | 7 | 2 | 6 | 0 | 6 | 7 | 4 | 4 | 8 | 0 | 0 | 0 | 0 | 4 | 8 | 0 | 7 | 0 | 7 | 8 | 0 | 8 | 2 |
| *Lutjanus argentiventris* | 8 | 8 | 7 | 1 | 7 | 7 | 7 | 7 | 4 | 4 | 6 | 7 | 8 | 5 | 8 | 4 | 5 | 0 | 8 | 7 | 5 | 8 | 8 | 8 | 8 | 7 |
| *Lutjanus colorado* | 0 | 0 | 0 | 0 | 0 | 1 | 0 | 0 | 3 | 0 | 0 | 0 | 5 | 0 | 2 | 0 | 1 | 0 | 0 | 7 | 0 | 0 | 0 | 7 | 2 | 5 |
| *Hoplopagrus guntherii* | 4 | 4 | 4 | 0 | 4 | 0 | 4 | 2 | 0 | 0 | 0 | 2 | 5 | 0 | 1 | 3 | 3 | 1 | 2 | 3 | 2 | 1 | 4 | 1 | 3 | 3 |
| *Lutjanus novemfasciatus* | 0 | 5 | 3 | 0 | 0 | 8 | 8 | 0 | 1 | 0 | 0 | 5 | 0 | 0 | 0 | 0 | 0 | 0 | 1 | 4 | 1 | 1 | 1 | 0 | 0 | 0 |
| *Lutjanus peru* | 8 | 2 | 0 | 7 | 8 | 0 | 0 | 7 | 6 | 6 | 7 | 0 | 0 | 0 | 0 | 0 | 0 | 8 | 0 | 0 | 0 | 1 | 8 | 2 | 3 | 0 |
| *Mycteroperca jordani* | nd | nd | nd | 3 | 2 | 3 | 0 | 2 | 0 | 1 | 0 | 0 | 0 | 0 | 0 | 0 | 1 | 0 | 0 | 0 | 0 | 1 | 0 | 0 | 0 | 0 |
| *Mycteroperca rosacea* | nd | nd | nd | nd | 3 | 0 | 0 | 4 | 3 | 3 | 1 | 2 | 3 | 3 | 2 | 2 | 3 | 3 | 2 | 3 | 3 | 1 | 4 | 2 | 4 | 0 |
| *Paranthias colonus* | 8 | 8 | 8 | 8 | 8 | 8 | 8 | 8 | 8 | 8 | 8 | 8 | 8 | 8 | 8 | 8 | 7 | 8 | 8 | 8 | 8 | 8 | 8 | 8 | 0 | 7 |
| *Seriola lalandi* | 0 | 0 | 0 | 4 | 3 | 0 | 0 | 3 | 0 | 1 | 0 | 0 | 0 | 0 | 0 | 4 | 3 | 0 | 0 | 0 | 4 | 0 | 0 | 0 | 1 | 0 |
| *Seriola rivoliana* | 4 | 1 | 0 | 4 | 0 | 2 | 0 | 4 | 3 | 2 | 1 | 0 | 0 | 0 | 0 | 0 | 2 | 4 | 0 | 0 | 2 | 2 | 1 | 1 | 0 | 0 |
| *Sphyrna lewini* | 0 | 6 | 3 | 2 | 0 | 0 | 0 | 4 | 0 | 1 | 2 | 5 | 5 | 3 | 6 | 5 | 3 | 1 | 4 | 6 | 0 | 0 | 0 | 0 | 0 | 4 |
| Number of replicates | 2 | 2 | 2 | 2 | 2 | 1 | 1 | 4 | 2 | 4 | 2 | 2 | 2 | 3 | 2 | 2 | 2 | 2 | 1 | 2 | 2 | 1 | 1 | 2 | 2 | 2 |

**Note:**
nd, indicates no data.

during three months, October 2003, May 2004, and June 2004, and a value was estimated for these months by averaging temperatures of the respective months collected during all other years (n = 5–6).

To further understand temporal patterns in the co-occurrence of seamount species, we plotted species abundance curves over time. This qualitative gradient analysis facilitated visualizing the seasonal separation and overlap of species at the seamount. We summarized census data from all five years into a single seasonal cycle (12 months) by taking the mean of all the census scores for each species by month, then normalizing by the highest value, so that relative abundance could be compared among species. We then fit a curve though the monthly values for each species using locally weighted smoothing via least squares quadratic polynomial fitting (Loess fit; Matlab$^{TM}$, Mathworks).

To compare the number of species present at the seamount with sea temperature, the presence or absence of the 17 species was summarized by month. A species was scored as present if it was observed during that month in any year during the study. An index of species richness for each month was determined as the sum of the number of target species that were present. The temperature gradient near the surface (SST–T30) during each month was calculated as the mean difference between SST and T30 over all years. We used standard linear regression analysis to determine whether there was a significant correlation between the number of species observed and SST, T30, or near-surface temperature gradient.

# RESULTS

## Oceanographic environment

The seamount environment was characterized by substantial variation in temperature both temporally and in terms of vertical thermal gradient. However, the annual cyclic pattern was relatively predictable. Between 1999 and 2005 the mean two-day Sea Surface Temperature at EBES was 24.4 °C and ranged from 18.3–32.5 °C (Fig. 1B). Sea surface temperature generally peaked near 32 °C in August and September then decreased rapidly from October through January to a low near 19 °C in February and March. Throughout summer, STT increased steadily but gradually. Mean daily T30 was 21.8 °C and ranged from 16.1–29.8 °C with maxima during September and October and minima from January to June. Colder deep water persisted during the spring and early summer as surface temperatures warmed, resulting in a strong and shallow thermocline that lasted through July. T30 then typically rose sharply to its yearly high peak. By November both SST and T30 typically fell rapidly as mixing deepened.

Periodic temperature profile measurements at EBES revealed the extent of seasonal changes throughout the water column. Most annual variability occurred above 200 m (Fig. 2). Temperature was relatively constant (~13 °C) at 200 m depth but there were considerable differences in the depth of the mixed layer and the degree of stratification in the upper near-surface layer. Although SSTs were similar in June and November (~27 °C) the temperature cast from June 1999 revealed strong stratification with a gradient of > 8 °C in the top 30 m. This observation is consistent with the average observed difference between mean SST and T30 in June (mean = 8.4), as shown by the difference between the

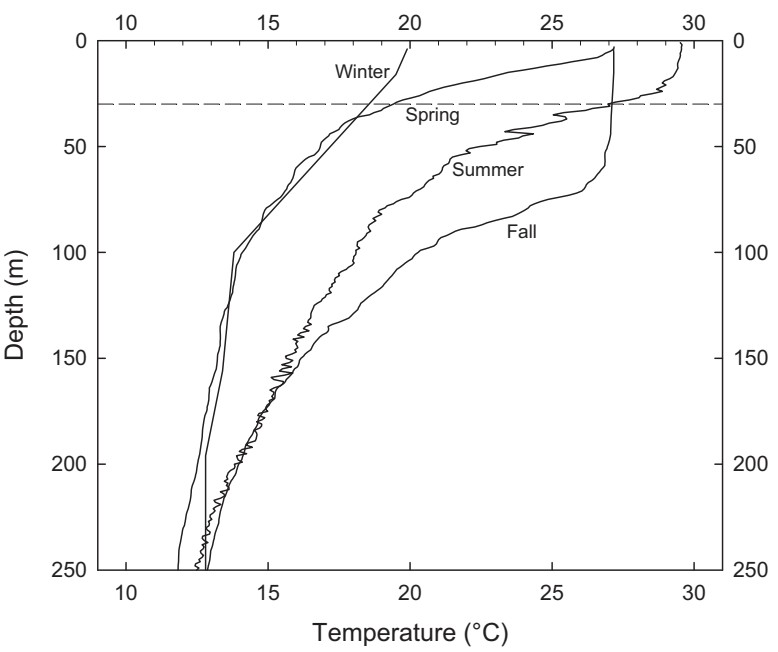

**Figure 2 Representative water column profiles of temperature immediately adjacent to EBES seamount.** Stratification above 30 m (dashed line) ranged from strong in June to very weak in November.

gray line and dark line each June in Fig. 1C. The cast from November revealed a deep mixed layer extending to ~70 m. This deep mixing generally persisted through December.

## Censuses

A total of 53 individual census dives were conducted at irregular intervals over 62 months from May 1999 to August 2004. These were averaged by calendar month resulting in 26 independent observations over the study period (Table 3). The relative abundance of species encountered during censuses at EBES at different times of the year revealed changes in the community structure. For many species, a clear seasonal signal was evident when relative abundance was plotted over time against SST and T30. For example, *Seriola lalandi* was generally absent from the seamount during warm seasons, but generally peaked in abundance from January to April, when temperatures were low (~20 °C; Fig. 3A). In contrast, peaks in *Lutjanus novemfasciatus* abundance coincided with positive peaks in SST and T30 between August and November when temperatures were high (~27 and 32 °C, respectively; Fig. 3B).

Sea surface temperature and T30 were both significant predictors (multiple logistic regression, $P < 0.05$) of seasonal relative abundance for eight species (Table S1). There was also a significant nominal (presence or absence) response ($P < 0.05$) to STT and T30 for *C. hippurus*, based on fishing surveys (Table S1). To better understand the differing responses to oceanographic cycles by community members, we used the coefficients of the two regressors ($\beta_{SST}$ and $\beta_{T30}$) as ordination axes and plotted each species in relation to these environmental gradients (Fig. 4; Table S1). For negative log-likelihood, a negative $\beta$ values along either axis denotes a positive effect, and a positive $\beta$ value indicates the

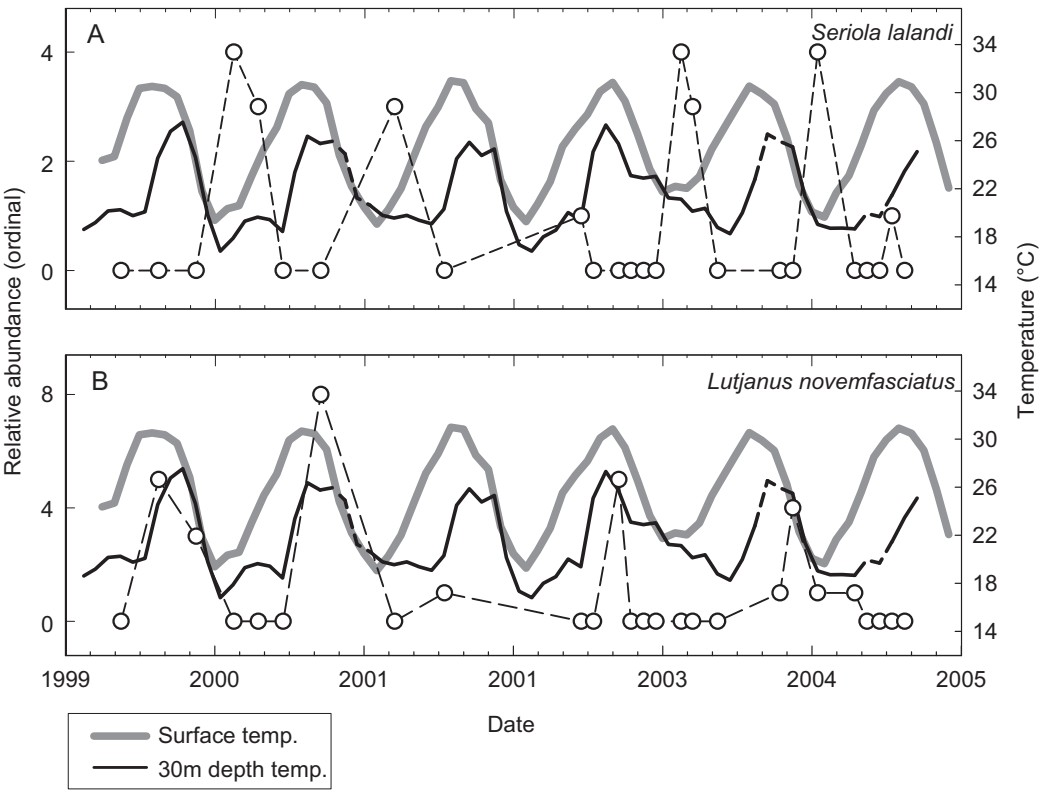

**Figure 3 Relative abundance of representative species.** Illustrative examples from the spring (A) *Seriola lalandi* and fall (B) *Lutjanus novemfasciatus* assemblages of fishes relative to mean monthly SST (gray line) and T30 (dark line) at EBES seamount from 1999–2005. Note that broken line portions in the T30 series represent estimated values for missing data.

opposite. Therefore, higher positive values for a combination of $\beta_{SST}$ and $\beta_{T30}$ values indicated a greater affinity with colder sea temperatures, and vice versa. The species fell into one of two groups divided by a line separating warm and cold affinity with slope = −1, and intercepts (0, 0), and we categorized individuals as cold or warm associated community members accordingly.

The cold associated group consisted of *L. peru*, *S. lalandi*, *S. revoliana*, and *M. jordani*. The warm associated group consisted of *C. caballus*, *C. hippurus*, *L. argentriventris*, *L. novemfasciatus* and *S. lewini*. A third group was represented year-round, as indicated by a non-significant negative log likelihood (P > 0.05). These species consisted of *A. solandri*, *D. macarellus*, *E. lineatus*, *H.guntherii*, *L. colorado*) *M. rosacea*, *P. colonus*, and *T. albacares*. The composition of species at EBES clearly differed seasonally, and we refer to three distinctive groups as the 'spring assemblage' (cold), 'fall assemblage' (warm), and 'year-round assemblage' (no significant response to temperature). We refer collectively to all three groups as the seamount-associated community.

## Species curves

To better visualize the seasonal process of species turnover we plotted species seasonal distribution curves (Fig. 5). The resulting curves indicate strong temporal partitioning among the fall and spring assemblages, with some overlap near the distribution tails at

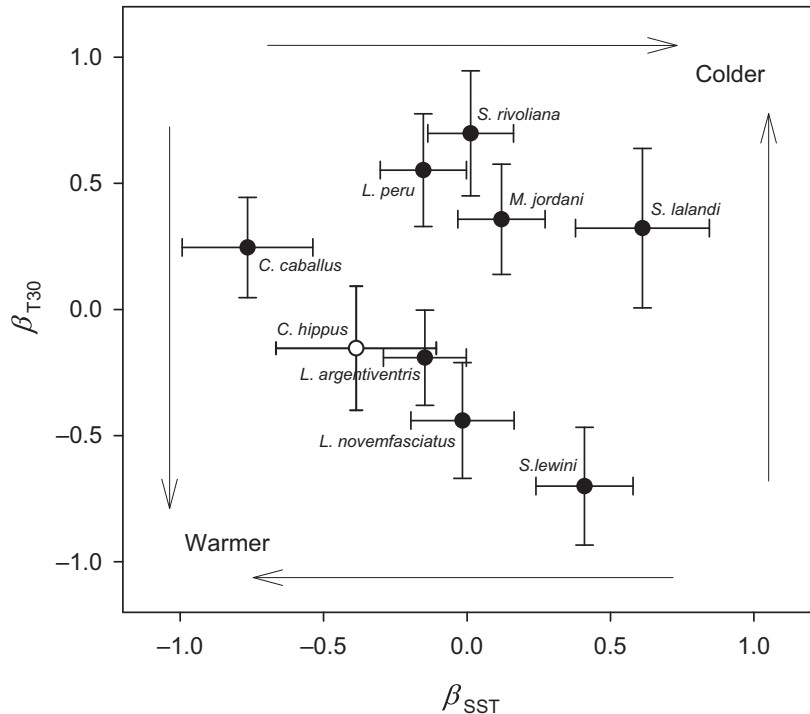

**Figure 4 Ordination analysis of seamount fish species assemblages.** Negative log likelihood parameter estimates of the two regressors, sea surface temperature (SST) and temperature at 30 m (T30), from logistic regression plotted on the plane $\gamma = \rho + \beta + SST + \beta T30$ revealed a spring (upper right) and fall (lower left) assemblage. Error bars represent standard error. Dark circles resulted from ordinal regression, and open circle resulted from nominal regression.

the boundaries that occurred during December through February and May through July (Figs. 5A and 5B). Within assemblages the curves were somewhat offset, but more similar within the spring assemblage than the fall. Abundance was near zero during warm months for all four spring species (also see Table 4), and the seasonal peaks where more aligned except that *S. lalandi* was shifted slightly earlier than the others. Among fall assemblage members, abundance curves did not overlap as clearly. Some members were absent altogether from the seamount during the colder months, while others were still present in low numbers. For example, *C. caballus* and *C. hippurus* were never observed in colder months between January through April and May respectively, while *L. argentriventris* was observed during every month of the year, but peaked in abundance during the warmer periods (see Table 4). Within the year-round assemblage abundance curves were generally less defined in amplitude and peaked at different times (Fig. 5C). Some notable exceptions include *L. Colorado* and *A. solandri* whose curves resemble those of the fall assemblage, and *M. rosacea*, which resembled the spring assemblage.

## Variation in species richness

The pattern of seasonal species turnover lead to variation in the cumulative number species observed during each month (min = 8 and max = 16 species). There was a positive linear relationship (P = 0.004) between the number of species and the surface to 30 m

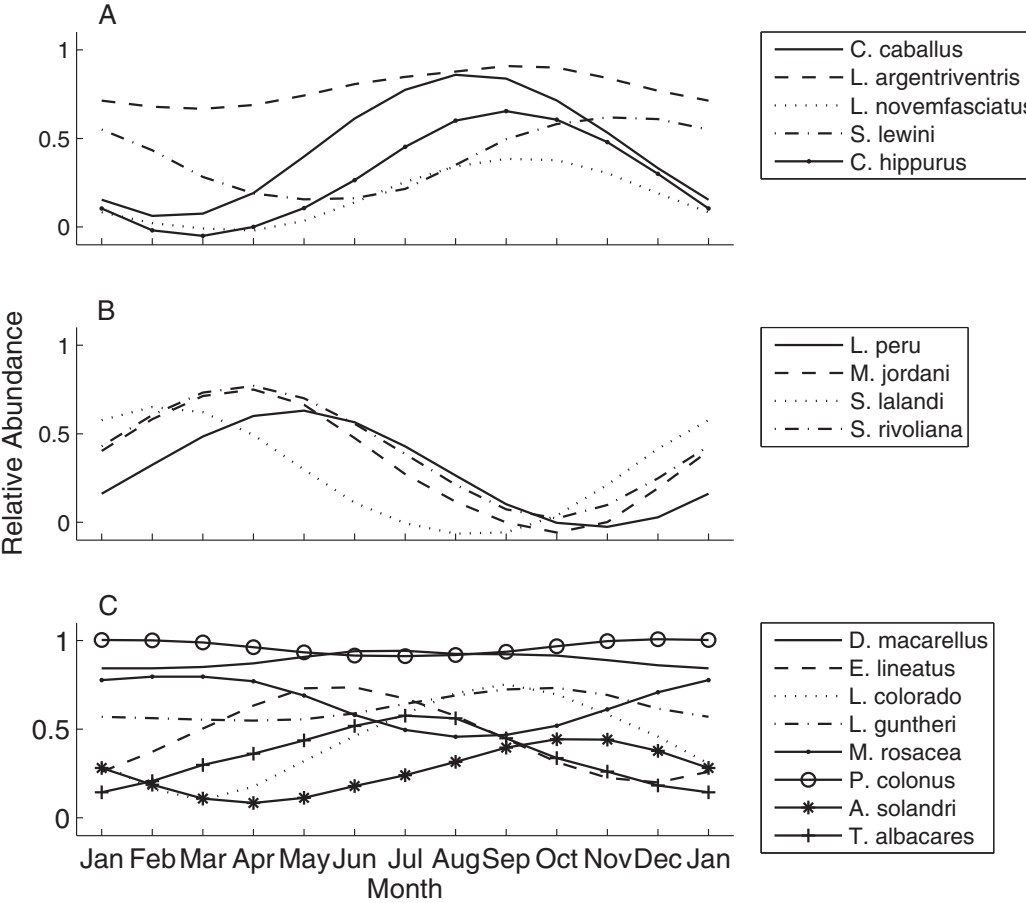

**Figure 5 Species abundance curves over time.** These qualitatively illustrate the occurrence of seasonal peaks in relative abundance of various fish species at EBES seamount. These species comprised the (A) fall, (B) spring and (C) year-round assemblages, respectively grouped based on logistic regression analysis output.

thermal gradient (Fig. 6). There was no significant relationship found between number of species and SST ($P = 0.139$) or T30 ($P = 0.921$). More species were observed when members of all assemblages overlapped from June through August, a season when greater thermal heterogeneity, warm surface and cool deep water ($\geq 30$ m), characterized the seamount environment (see Fig. 1C).

## DISCUSSION

UVCs are particularly challenging in open water where large schools of rapidly swimming individuals need to be quantified and recorded. The method of log scale ordinal bins presented here provides a simple and accurate way to estimate relative abundance where precision is scaled inversely with sampling difficulty to minimize observer error. The resulting ordinal binning is appropriate for logistic regression; a robust multivariate approach relevant to zero-inflated datasets. The resulting patterns of variation in the composition and number of species provided insights relevant to the processes of shallow seamount fish assemblage. Although the study was confined to a single seamount,

**Table 4 Presence or absence of species from both dive and fishing surveys observed by month.** Species were given a positive score (●) if they were ever observed during the month over the five-year study period.

| | Month | | | | | | | | | | | |
|---|---|---|---|---|---|---|---|---|---|---|---|---|
| | 1 | 2 | 3 | 4 | 5 | 6 | 7 | 8 | 9 | 10 | 11 | 12 |
| **Fall** | | | | | | | | | | | | |
| *Caranx caballus* | | | | | ● | ● | ● | ● | ● | ● | ● | ● |
| *Coryphaena hippurus* | | | | | | ● | | ● | ● | ● | ● | |
| *Lutjanus argentriventris* | ● | ● | ● | ● | ● | ● | ● | ● | ● | ● | ● | ● |
| *Lutjanus novemfasciatus* | ● | | | ● | | | ● | ● | ● | ● | ● | |
| *Sphyrna lewini* | | ● | ● | | ● | ● | ● | ● | | ● | ● | ● |
| **Spring** | | | | | | | | | | | | |
| *Lutjanus peru* | | ● | ● | ● | ● | ● | ● | ● | | | | |
| *Mycteroperca jordani* | | ● | ● | ● | | ● | ● | | | | | |
| *Seriola lalandi* | ● | ● | ● | ● | | ● | ● | | | | | |
| *Seriola rivoliana* | ● | ● | ● | ● | ● | ● | ● | ● | | | | |
| **Year-round** | | | | | | | | | | | | |
| *Acanthocybium solandri* | | | | | | ● | ● | | ● | ● | ● | ● |
| *Decapterus macarellus* | ● | ● | ● | ● | ● | ● | ● | ● | ● | ● | ● | ● |
| *Euthynnus lineatus* | | ● | ● | ● | ● | ● | ● | ● | ● | | ● | |
| *Hoplopagrus guntherii* | ● | ● | ● | ● | ● | ● | ● | ● | ● | ● | ● | ● |
| *Lutjanus colorado* | | ● | | | | ● | ● | ● | | ● | ● | ● |
| *Mycteroperca rosacea* | ● | ● | ● | ● | ● | ● | ● | | ● | ● | ● | ● |
| *Paranthias colonus* | ● | ● | ● | ● | ● | ● | ● | ● | ● | ● | ● | ● |
| *Thunnus albacares* | | ● | | | ● | ● | | ● | ● | | | |

the patterns observed over a five-year period were clear and can be evaluated in the context of potential hypotheses explaining elevated predator richness at seamounts.

## Functional similarities in distinct fish assemblages

For species that were present year-round, the seamount environment likely fulfills multiple vital life-history functions, including feeding, refuge, and reproduction. However, what specific function does the seamount provide for the spring and fall assemblages? The arrival of certain community members at EBES coincided with reported months of spawning aggregations in the Gulf of California. These included members of both the fall and spring groups. *L. novemfasciatus* generally peaked at EBES near September (see Fig. 3B), which is the reported spawning season for this species in the Gulf of California (*Sala et al., 2003*). Apart from these peaks in abundance, *L. novemfasciatus* generally was absent from EBES. *Sala et al. (2003)* also reported *S. lalandi* aggregations spawning at reefs and seamounts in April, citing observations of high densities and hydrated eggs in gonads. Peak abundance of *S. lalandi* at EBES occurred between January and April (see Fig. 3A). All three female *S. lalandi* sacrificially sampled in the fishing

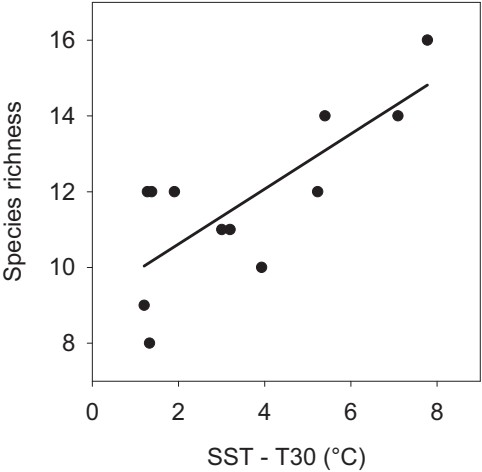

**Figure 6 Linear relationship between number of species and vertical thermal gradient at EBES seamount.** The number of observed species increased in months with greater stratification (n = 12 months, $r^2$ = 0.587, P = 0.004).

surveys at EBES during a large aggregation on February 3, 2003 had large gonads with hydrated eggs. A single male captured during this event released milt as it was brought aboard. Thus the arrival of *L. novemfasciatus* and *S. lalandi* at EBES, although during opposite seasons, were both linked to yearly spawning (*Sala et al., 2003*). Migrations to site and season-specific spawning aggregations, can be on the order of hundreds of kilometers (*Bolden, 2000*), could potentially be to find conspecifics or to select environmental conditions favoring larval survival (*Johannes, 1978*).

Like many seamounts, EBES is a distinct geophysical feature, and is likely to function as a navigational (reference) or meeting (destination) point where wide-ranging organisms come together for mating events or otherwise. *Sphyrna lewini*, a member of the fall assemblage, aggregates in large groups at EBES and other seamounts (*Klimley & Nelson, 1984*). Previous studies have illustrated their emigration, alone or in small groups, away from EBES to surrounding open waters at night, presumably to forage, and return to the seamount the following dawn (*Klimley & Nelson, 1984*; *Klimley, 1993*). While foraging and mating apparently do not occur at EBES, during the day, the dense schools of hammerhead sharks remained closely associated with EBES slowly swimming back and forth along the ridge. Thus seamounts appear to provide a refuge and spatial reference.

Enriched foraging potential at shallow seamounts is another potential attraction for visitors and may increase their residence time. *Fonteneau (1991)* suggested that certain seamounts may enhance foraging by *T. albacares,* and some, but not all seamounts are associated with increased catch per unit effort of *T. albacares* and other tuna species (*Morato et al., 2010b*). At EBES schools of *T. albacares* were frequently observed feeding (S. Jorgensen, 2005, personal observation). Individual tunas tagged with acoustic tags remained resident near EBES at all times of the year for periods ranging from a few days to greater than a year (*Klimley et al., 2003*). During prolonged residence, the intervals between detections were brief and indicated the tunas could rarely have moved more than

900 m from EBES. While not all tagged *T. albacares* resided at the seamount, those that did likely foraged in close association with EBES over extended periods. In addition to foraging, seamounts also may play a role in navigation for *T. albacares* (*Holland, Kleiber & Kajiura, 1999*), which could explain why some tuna remained resident while others did not (*Klimley et al., 2003*).

From these observations of just some of the assemblage members, it seems clear that pelagic and reef-associated fishes (see Table 1) co-occur at EBES to fulfill a variety of functions including foraging, spawning and navigation. These and possibly other drivers attracted seamount visitors during a range of seasonally varying environmental conditions. There was a large range in the overall temperature and water column thermal structure (Figs. 1 and 2). Yet spawning, foraging, and navigation occurred at various times throughout the year (e.g. spawning by fall and spring assemblage members) despite the contrasting environmental conditions.

## Assemblage cohesion

Our data reveal how local seamount communities are ephemeral and structurally dependent on seasonal and regional oceanographic conditions. At EBES distinct assemblages were associated with pronounced seasonal changes in oceanography. Many pelagic fish species, such as tunas, track thermal fronts and other oceanographic features (*Laurs, Yuen & Johnson, 1977*; *Kitagawa et al., 2007*; *Schaefer, Fuller & Block, 2007*) and reside for extended periods around seamounts (*Holland, Kleiber & Kajiura, 1999*; *Klimley et al., 2003*). Thus, the movement of oceanographic features over seamounts may determine the pool of pelagic species available at a given time and the composition of seamount-associated communities.

*Klimley & Butler (1988)* hypothesized that some fish species groups may occur as 'mobile communities' in the open ocean arriving and departing from local seamounts as a single unit. We were unable to track the precise arrival and departure timing of assemblage members; however, a number of results suggest that members converged at EBES under slightly different conditions, and for different purposes. Ordination analysis clearly grouped overall warm associated species distinctly from the cold associated group (see Fig. 4). Within these groups, however, further structure was evident. For example, in the fall assemblage, *C. caballus* abundance was highly correlated with warmer SST, but also negatively correlated with T30 (Fig. 4; Table S1); conditions that prevail in summer. In contrast *S. lewini* abundance was highly positively correlated with warmer T30, but negatively correlated with warmer SST (Fig. 4; Table S1); typical fall conditions (Figs. 1 and 2). This structure was further evident in the offset of abundance curve peaks between *C. caballus* in August, and *S. lewini* in November (Fig. 5A). The remaining fall assemblage members peaked in abundance somewhere between these two (Figs. 4 and 5). Similar structure, although more subtle, occurred for the spring assemblage.

This general offset in peak relative abundance suggests that members likely did not arrive together. Furthermore, within the fall assemblage, some species were present at EBES in low abundance year-round during the off-peak seasons (e.g. *L. argentiventris*) while others were completely absent when not in peak abundance (e.g. *C. caballus*).

Finally, within-assemblage differences in the nature of each species association with the seamount (e.g *L. novemfasciatus* gathered to spawn, while *S. lewini* aggregated at EBES during daylight and foraged away at night) suggest that at least some members co-occurred at the seamount for dissimilar reasons. We conclude that assembly of seamount community members at EBES was largely the result of 'species-individualistic' processes. The associations between aggregating species were seasonally ephemeral, and likely confined to this place and time. Once aggregated at the seamount, however, it seems possible that sudden movements of water masses, on short time scales, might influence an entire assemblage, away from or back to a seamount (*Klimley & Butler, 1988*).

## Species richness

Water temperature is a strong predictor for species richness among broad marine taxa; for open ocean fishes and foraminifera this convex function peaks near 25 °C (*Rutherford, D'Hondt & Prell, 1999*; *Worm et al., 2005*; *Whitehead, McGill & Worm, 2008*) and at corresponding intermediate latitudes (*Morato et al., 2010a*). The mean SST at EBES was 25.4 °C and ranged seasonally from 18–30 °C. Given the movement potential of the seamount-associated species and the prevalence of seasonally advancing and retreating thermal fronts (*Klimley & Butler, 1988*; *Trasviña-Castro et al., 2003*; *Douglas et al., 2007*; Fig. 1), one might predict that the greatest number of species would occur at EBES when temperature were near 25 °C. A mean SST of 25 °C occurred twice each year at EBES; during summer (May) and late fall (November) transitions, respectively (Fig. 1). However, increased predator richness at the seamount was not predictable by surface temperature alone. Instead, it was correlated with greater vertical thermal gradient. Interestingly, the strongest gradients occurred in summer and the weakest in late fall. In essence increased species richness did occur when surface temperature was near 25 °C, but only during summer when additional processes occurring below the surface where the likely drivers.

The early summer peak in total species numbers at EBES was largely a transition period after the average peak in abundance of the spring assemblage (March) and before that of the fall assemblage (September); apparently not the prime 'target' period for either group. That species co-occurred for a number of different functions also suggests that it was not linked to a single prevalent environmental state. In fact the highest variability in temperature, both temporally and spatially (i.e. in the water column), occurred then, suggesting increased habitat heterogeneity itself is likely to be an important factor. Summer is a warming transition period at EBES when warm and cold water occurs simultaneously providing a thermally heterogeneous habitat. This range of temperatures, accessible across a narrow range of depths, may accommodate the thermal optima (*Blank et al., 2002*; *Boyce, Tittensor & Worm, 2008*) of a wider range of species thereby enabling the co-occurrence of both fall and spring assemblages. Globally, over broad spatial and temporal scales, pelagic tuna and billfish diversity positively correlates with spatial temperature (SST) gradient, and where data are available, tuna and billfish diversity correlates in turn with total predator diversity (*Worm et al., 2005*).

Our results suggest that increased habitat heterogeneity through vertical and temporal thermal gradients may be an important mechanism for increased species richness at the local scale and may also shape patterns of diversity of vagile ocean fishes at larger scales. The turnover of species at EBES, or 'beta diversity,' was clearly reflected through time with changes in temperature. In turn these species co-occurred (overlapped) when a wide range of thermal habitat was available.

In addition to habitat heterogeneity, a potential alternative explanation is that enhanced foraging opportunities may increase species richness via the 'species-energy' hypothesis (*Worm, Lotze & Myers, 2003*; *McClain, 2007*; *Morato et al., 2010a*). What can seasonal variation at EBES tell us about a trophic link to observed changes in species richness? Local food supply could be augmented, either through vertical mixing and elevated primary productivity, or through concentration of macroplankton (*Wolanski & Hamner, 1988*; *Genin, 2004*). It is unlikely that the residence time of upwelled water around seamounts is sufficiently long for primary production enrichment to propagate up the food web to the level of predatory fishes (*Genin, 2004*). However, a match between enhanced primary productivity and early life-history survival (*Cushing, 1990*) could increase larvae fitness for fishes spawning at seamounts whether water masses (including nascent larvae) were advected or retained. That at least two species aggregated at EBES to spawn suggests enhanced primary production at seamounts could be one mechanism for increasing predator fish richness that is not directly related to the 'species-energy' model.

Primary productivity in the Gulf of California is typically highest during fall/winter, when strong winds drive regional upwelling, and lowest in spring/summer (*Douglas et al., 2007*). However, another important mechanism for primary productivity in the Gulf is tidally driven vertical mixing (*Trasviña-Castro et al., 2003*; *Douglas et al., 2007*). At EBES dynamic instability and mixing due to vertical current sheer occur over the top of the seamount (*Trasviña-Castro et al., 2003*; *Douglas et al., 2007*). Vertical mixing at this depth would likely be drowned out in fall/winter when the mixed layer typically extends well below 50 m (Fig. 2) but may become locally important during spring/summer when the thermocline is shallow and wind driven production is regionally limiting.

The most widely accepted mechanism for enhanced foraging at seamounts involves a biophysical match between currents and animal behavior (*Genin, 2004*; *McClain, 2007*). However, quantifying prey (zooplankton) availability was beyond the scope of this study. While this mechanism cannot therefore be directly addressed as a contributing factor for increased richness, two observations suggest it may not be the primary driver at EBES. First, many species arrived at the seamount for reasons other than foraging. Second, the summer period, when the greatest number of species was observed, appeared to be a 'transition' period rather than a 'target' period for either fall or spring assemblages. If foraging were the primary common driver explaining increased richness during summer, then one might expect more species to coincide in their peak abundance.

Ultimately these observations are also more consistent with the 'habitat heterogeneity' hypothesis since fish species occurred at EBES for a variety of functions including

foraging, spatial reference and reproduction during different seasons. The intra-annual turnover of species, or temporal beta-diversity resulted in seasonal variation in fish species numbers at EBES that was positively correlated with thermal heterogeneity in the water column. Although this study was limited to a single seamount location, the UVC methods and clear patterns observed enable comparisons to other locations, and in this case provide support for the idea that oceanographically heterogeneous and dynamic seamounts may support greater predator richness.

## ACKNOWLEDGEMENTS

Considerable logistical support was provided by H. Fastenau, L. Inman, F. McLeese, M. Cota, J. Richert, J. Downs, I. Nevius, M. Silva, A. Trasviña-Castro, and numerous staff of Centro de Invesigaciones Biologicas del Baja Norte of La Paz, and the Cortez Club. We also thank C. Logan for helpful review and comments on the manuscript.

### Funding

This work was funded by the Biological Oceanography Program of the National Science Foundation (grant: OCE-9802058), UC MEXUS, the Fulbright Association, and CONACyT of Mexico (grant: PN-9509-1995 and PN-1297-1998). The funders had no role in study design, data collection and analysis, decision to publish, or preparation of the manuscript.

### Grant Disclosures

The following grant information was disclosed by the authors:
Biological Oceanography Program of the National Science Foundation: OCE-9802058.
UC MEXUS, the Fulbright Association, and CONACyT of Mexico: PN-9509-1995 and PN-1297-1998.

### Competing Interests

The authors declare that they have no competing interests.

### Author Contributions

- Salvador J. Jorgensen conceived and designed the experiments, performed the experiments, analyzed the data, contributed reagents/materials/analysis tools, wrote the paper, prepared figures and/or tables, reviewed drafts of the paper.
- A. Peter Klimley conceived and designed the experiments, performed the experiments, contributed reagents/materials/analysis tools, reviewed drafts of the paper.
- Arturo Muhlia-Melo conceived and designed the experiments, performed the experiments, contributed reagents/materials/analysis tools, reviewed drafts of the paper.
- Steven G. Morgan contributed reagents/materials/analysis tools, reviewed drafts of the paper.

## Animal Ethics

The following information was supplied relating to ethical approvals (i.e., approving body and any reference numbers):

Ethical review was not required for sampling of marine fishes in fishing surveys.

## Data Deposition

The raw data has been supplied as Supplemental Dataset Files.

## Supplemental Information

Supplemental information for this article can be found online at http://dx.doi.org/10.7717/peerj.2357#supplemental-information.

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
