# Peer review of "Seasonal changes in fish assemblage structure at a shallow seamount in the Gulf of California"

_PeerJ, doi:10.7717/peerj.2357_

## Round 0.1 · original submission · Minor Revisions

· Academic Editor

Minor Revisions

I am sorry for the delay in getting this back to you. A third referee kept promising to get us back their review, but has not been able to do so, and rather than delay any longer, I have decided to move forward with the two reviews currently in hand. Although both referees were enthusiastic about the value of the manuscript and the work, both each had a number of suggestions for improvement throughout and attached a marked-up manuscript for your consideration. The suggested changes are largely editorial in nature and will likely clarify the text for future readers, but I see nothing that is likely difficult for you to address. I look forward to reading your revised manuscript which addresses these referee comments.

Reviewer 1 ·

Basic reporting

The article is well written and in my opinion the manuscript is in line with the standards of the PeerJ.
The structure is appropriate the Tables and Figures are also appropriate and sufficient
The article is interesting and well supported in a good data series. Raw data is made available by the authors.
Authors were ambitious in trying to answer or explain the changes on the species richness over seamounts and how they related with two main hypothesis authors describe as the "species-energy" hypothesis and "habitat heterogeneity" hypothesis. They recognize however that they would need more that (for example regarding the productivity, zooplankton availability, or a better quantification of prey species in general) to better understand the possible mechanisms that could explain the predators differential occurrence along the year. This would help to better clarify the causes or the mechanisms behind those changes.

Experimental design

The research questions were well defined and the data collected has good quality. We can imagine the difficulties involved to undertake this study, both due to the numbers of years it take as to the nature of the environmental where it takes place. Underwater visual censuses over a shallow seamount is in fact challenging and maintain the study during several years is undoubtedly. something uncommon and relevant.
The study and the scientific questions it try to answer are highly relevant to the seamount science and the study is in this sense somehow original. The study give important information about the occurrence and changes on the composition of fish species over shallow seamounts and give sounding explanations for the mechanisms that may explain these variability or changes.
I would like to see some comment regarding the number of divers involved and how comparable are their counts. Is there any previous calibration among divers? I imagine they are experienced divers but for a study which take about 5 years, it would be good have more information. The divers were always the same?
The experimental design is sounding and the statistical analysis appropriate.
I do not find any ethical issues regarding this study.

Validity of the findings

The statistical analysis was sounding and appropriate. Authors present the statistical results on a Supplemental Table. The replication level is not high in some months but this do not compromise the analysis because the monthly data was grouped from all years increasing the monthly number of observations and giving consistency to the findings.
The study introduces some novelty to the seamount ecology studies and have impact over a large audience. It could also be important for seamount management purposes.
The study suggests some causes to the variation on species composition along the year on the seamount which are not completed verified. This is the case for the sentence “select environmental conditions favoring larval survival” (line 355) to justify why the species aggregate over the seamount during the spawning season. It could be the case that most of the larvae do not survive over this isolated seamounts and this speculation should be clarified.
The study lacks any type of attempt to quantify the possible prey species and this could be helpful in clarifying the possible “enriched foraging potential” (line 368), or the increase of the primary productivity and its influence on the spawning behavior of the species.
However the study open the door for further studies which could try to answer to this speculations. The oceanographic heterogeneity hypothesis is sounding in explaining the predator richness but it seems that the more research should be done to find if other primary or more basic causes can explain this or if it is just a consequence of the fact that wider environmental conditions (heterogeneity) may attract species with different optimal habitat conditions simultaneously (ecotone or the frontier effect).

Additional comments

The research questions were well defined and the data collected has good quality. We can imagine the difficulties involved to undertake this study, both due to the numbers of years it take as to the nature of the environmental where it takes place. Underwater visual censuses over a shallow seamount is in fact challenging and maintain the study during several years is undoubtedly. something uncommon and relevant.
The study and the scientific questions it try to answer are highly relevant to the seamount science and the study is in this sense somehow original. The study give important information about the occurrence and changes on the composition of fish species over shallow seamounts and give sounding explanations for the mechanisms that may explain these variability or changes.
The study lacks any type of attempt to quantify the possible prey species and this could be helpful in clarifying the possible “enriched foraging potential” (line 368), or the increase of the primary productivity and its influence on the spawning behavior of the species.
However the study open the door for further studies which could try to answer to this speculations. The oceanographic heterogeneity hypothesis is sounding in explaining the predator richness but it seems that the more research should be done to find if other primary or more basic causes can explain this or if it is just a consequence of the fact that wider environmental conditions (heterogeneity) may attract species with different optimal habitat conditions simultaneously (ecotone or the frontier effect).

I would like to see some comment regarding the number of divers involved and how comparable are their counts. Is there any previous calibration among divers? I imagine they are experienced divers but for a study which take about 5 years, it would be good have more information. The divers were always the same? How this could affect the results?

Annotated reviews are not available for download in order to protect the identity of reviewers who chose to remain anonymous.

Reviewer 2 ·

Basic reporting

Clear language was used throughout. There are a few instances identified in the annotated PDF where sentences and/or phrases could be rephrased to enhance clarity. Intro and background section were well written and relevant. It may be appropriate to include a reference to the seamount ecology book (Pitcher et al 2007). I disagree with your point that deeper seamounts are better studied than shallow seamounts. I think the literature is clearly skewed to research on seamounts whose summits are shallower than 500 m. Therefore, I think you misconstrue the current state of knowledge on deep seamount habitats. I do acknowledge that relatively few studies look at extremely shallow seamounts such as those with depths within SCUBA range, and I agree that this should be highlighted in your introduction. This is a unique and very interesting dataset.
The structure of the paper conforms to the standard and is clear and easy to follow.
Figure 1 is very informative and well done. However, it might be nice to have a more detailed topographic view of the seamount itself, instead of only including the larger geographic location of the seamount. Figure 4 is a bit confusing. Defining BTT in the figure label would help (I assume it stands for bottom temperature). Why is one species location the result of nominal regression when the rest used ordinal regression? Were the other non-temperature species removed for clarity? Did they fall on a middle line? Could they be included perhaps with smaller symbols?
I have included detailed comments on Figure 5 in the annotations of the pdf; however, in summary, I don’t quite understand how the species were split into summer, winter, and year around assemblages because based on the shape of the abundance curves many seem to be in the wrong plot (eg L colorado seems to have a much clearer summer response than L argentriventris), which is mentioned briefly in the text but never fully discussed. For Figure 6 I am not sure why you chose to model the month averaged data instead of using all of the raw species richness data which would give you much more than 12 data points for the regression. However, the figure is clear and well labeled.
In terms of the raw data, only the temperature data was supplied. I think it would be very useful to see the raw count data as well, especially because these were converted to an ordinal scale. A reader can go from the raw counts to the ordinal scale given by Table 2, but Table 3 alone does not provide count data. Perhaps Table 3 could be modified to show counts instead of ordinal values or an additional supplemental data table with the counts could be included. Table 4 is very useful, but the table description does not mention the source of the data. Is this presence/absence data from fishing or derived from the SCUBA surveys?

Experimental design

The authors clearly stated that their aim was to investigate the correlation between the thermal habitat at EBES and the pelagic fish assemblage. This question is interesting, relevant, and meaningful and addresses an identified knowledge gap: how are seamount communities influenced by oceanographic conditions. The data set was impressive with 5 years of monthly surveys. Using the difference between temperature at 30m and the surface was an interesting and clever way to quantify the amount of water stratification with the data available.
Methods were generally clear and well described. However, I am still not sure why the bins were organized such that classes were not defined by whole numbers. I understand the need to organize counts into expanding width bins having conducted fish surveys on SCUBA myself. However, why not just define classes by whole numbers (eg bin one 0-1 etc)?
For the methods for the environmental records I am uncertain as to why the shark tag data were described in the methods because they are not used later in the paper.
For the analysis multiple logistic regression which was appropriate to model the ordinal scores; however, in the analysis methods it states that variation in encounter rate was regressed over the various environmental data. What was the response variable in the model? Ordinal score or encounter rate or are these the same? Please define encounter rate and how it was calculated from the raw data/ordinal scores.

Validity of the findings

Conclusions are generally clearly stated and well supported. However, the groups determined by the ordination seem to be different than groups made from both the presence/absence fishing data and the relative abundance curves. How do you explain this discrepancy? (see comments on Figure 5).
I believe the general conclusion that thermal habitat influences the seamount community composition due to species specific preferences for specific thermal ranges is sound and well supported. However, because the data presented here show the importance of thermal habitat to these pelagic species, I am a bit unconvinced of the seamount link. Many of these species are wide ranging, and if water temperature is the most important factor in determining presence and abundance of these species, what does the seamount have to do with the story other than the data were taken at a seamount? There is no data presented in this paper that leads me to believe that these species are in the area specifically due to the seamount –rather I think this data shows more general seasonal differences in the pelagic fish community in the southern Gulf of California. An explicit link to the seamount would be needed with something like larger spatial scale catch data or a cited literature reference showing that these species are truly seamount associated/seamount aggregated in this region. The spawning aggregation argument is compelling, but I don’t think there is enough data to show that the some of the species counted here traveled to the seamount only to spawn. The connection to functional use of the seamount is a bit thin and not directly supported by the data presented in the study. From the data presented here I would expect to see similar thermal patterns for these same species at an entirely offshore pelagic site in the Gulf as well.
The connection between increased foraging opportunities at seamounts and the seasonal variability in community composition is interesting, but I think it is still difficult to disentangle seamount specific patterns from the general Gulf productivity pattern. The winter assemblage could be immigrating to the Gulf for the higher productivity regardless of the seamount, whereas the summer assemblage could be actively aggregating at the seamount because of relatively higher productivity there due to seamount mixing effects. However, this is difficult to show.
Overall the discussion was well written and thought provoking.

Additional comments

This is a very interesting and unique data set that further demonstrates the complexity of seamount ecosystems. Seamount ecology paradigms often try to paint an overly simplistic view of the seamount effect and try to sell the idea that seamounts are consistent biological hotspots. However, data such as these are crucial to break down this view and show that the seamount effect is complex, often species specific, and influenced by local oceanographic conditions. I think a still unfilled gap in this dataset is the comparison to an off seamount site, so that some comment can be made about the role of the seamount itself in the seasonal patterns presented here.

Annotated reviews are not available for download in order to protect the identity of reviewers who chose to remain anonymous.

---

## Round 0.2 · accepted · Accept

· Academic Editor

Accept

Thanks for your detailed response to the referee comments. I am satisfied with the revisions, and the referee comments were not major changes, so I see no reason to send it back to them. I feel that you have dealt with the comments sufficiently and am happy to move the manuscript forward into production at this time.